# 'I haven't met them, I don't have any trust in them. It just feels like a big unknown': a qualitative study exploring the determinants of consent to use Human Fertilisation and Embryology Authority registry data in research

Claire Carson,[1] Lisa Hinton,[2] Jenny Kurinczuk,[1] Maria Quigley[1]

[1]National Perinatal Epidemiology Unit, University of Oxford, Oxford, UK
[2]Health Experiences Research Group, Department of Primary Health Care Sciences, Oxford University, Oxford, UK

**Correspondence to**
Dr Claire Carson;
claire.carson@npeu.ox.ac.uk

## ABSTRACT

**Objectives** To explore why and how fertility patients decide to allow (or deny) the use of personal data held in the Human Fertilisation and Embryology Authority registry for linkage and research.

**Design** A qualitative study was conducted using in-depth face-to-face interviews and an online survey to garner information on experience and opinions from fertility clinic patients and staff. Verbatim transcripts were analysed using the 'one sheet of paper' method to identify themes.

**Setting** Women and men were recruited between September 2015 and December 2017, via fertility clinics across England and online advertising, then interviewed at a location convenient to them.

**Participants** 20 patients and 9 staff were interviewed, 40 patients completed the online survey.

**Results** Consent for disclosure (CD) forms are completed at a stressful time, when patients often feel overwhelmed; these forms were considered a low priority. Perceptions of benefit (to individuals, to wider society) and harm (misuse of data, impact of disclosure on child) influenced consent. Important themes included: understanding of the forms; trust in those asking, in researchers, in the Human Fertilisation and Embryology Authority (HFEA); and wider attitudes to data use. Issues influencing response, and thus the representativeness of the HFEA data set, were highlighted.

**Conclusions** Understanding what is being asked, and trust in those organisations keeping and using personal data, affects individual decisions to consent to disclosure. Patients were influenced by the wider context of infertility, as well as general concerns about data sharing and security. Low consent rates, which vary by clinic and likely also by patients' characteristics, have adverse implications for research conducted using HFEA data collected after 2008. Public understanding of data use and security is relatively poor; increased public trust in, and awareness of, research based on routine data could improve consent to data use and reduce the risk of bias.

## INTRODUCTION

Routinely collected health and administrative data record valuable information

### Strengths and limitations of this study

► Since 2009 the Human Fertilisation and Embryology Authority have sought consent from patients to allow patient data to be used for research and linkage. Consent rates were initially low (around 30%), and although they have increased (to around 70%) the reasons for agreeing or refusing to give consent to allow the use of personal identifiable data remain unknown.

► In-depth qualitative research methods afforded a richer data set than could be collected via a structured survey or questionnaire alone.

► Experiences were sought from a diverse group of patients and staff.

► Recruitment was challenging, as the study seeks to enrol those individuals who have previously declined to participate in research.

► Augmenting the interviews with anonymous online data collection was used to address this constraint.

about health behaviours, treatments and outcomes. When these data are combined through record linkage their full potential becomes clear: researchers can use existing, routinely collected large data sets to address questions that would otherwise be too expensive or time-consuming to investigate using primary data collection. Such linkage requires the use of unique identifiers and in the UK, where there is no national identity number, this necessitates access to personal identifiable information (PII) such as name, date of birth, postcode or National Health Service (NHS) number.

The use of patient data in research has come under increasing scrutiny in the UK since the mishandling of the care.data plans in 2013.[1] Care.data was intended to

make available linked primary and secondary care data for England to 'approved users' (including researchers and care commissioners) in pseudonymised data sets. A botched awareness campaign led to a public backlash about the lack of information, and concerns over data security and usage then put an end to what should have been an incredibly useful resource for health research and planning purposes. Subsequent high profile stories related to the use of personal data without consent have compounded the problem, including the release of data from more than 1 million NHS patients to Google's DeepMind[2] and the more recent harvesting of Facebook profiles for political purposes.[3] This has brought the issue of data ownership, use and control acutely into focus.

Efforts to inform and improve the debate about data use and the benefits of sharing and linking health data have had some impact (eg, the Wellcome Trust's Understanding Patient Data[4]) but, in general, understanding remains poor.[5] Many patients erroneously believe that their data are readily available to researchers.[6] Reported attitudes to data use in the UK reflect this confusion: while one recent survey of the general public reported that 83% feel health research is very important[7] and 77% support the use of their health data provided it is anonymised,[8] another survey found just 47% of respondents were prepared to share medical data when PII is included.[9] Few seem to recognise that PII is required for the record linkage that facilitates such research.

Gaining informed consent for record linkage studies is logistically challenging, and low consent rates have important implications for the representativeness of the study population, reducing the generalisability of research findings. For surveillance and prevalence estimates, complete information is needed, while for aetiological studies, bias (and therefore spurious results) may arise if those who consent to data use differ from the source population. These issues are recognised and have been addressed through legislation for some data sources. Due to their national importance and because asking every patient for permission is unworkable, cancer registries in England and Wales are permitted to record and process PII without consent. This use of PII and health data without consent due to legitimate public benefit also applies to other data sources, but not to the UK register of fertility treatment.

### Fertility data are a special case—the HFEA register

Approximately one in six couples in the UK have problems conceiving,[10 11] and assisted reproductive technologies (ARTs) such as in vitro fertilisation (IVF) are increasingly commonly used. As a result, more than 1 in 50 children are now born after ART.[12] Since 1991, legislation has required the regulation of ART and the recording of treatment in a registry maintained by the Human Fertilisation and Embryology Authority (HFEA). As one of the oldest and most complete registries of fertility treatment globally,[13] this represents an underexploited resource to investigate the long-term health

implications of ART. Consent for the use of these data by researchers was not initially sought. Under current UK law HFEA data recorded between 1991 and 2008 can be used for research purposes (including linkage) without consent, provided that the societal benefit outweighs the potential for individual harm. However a legislative amendment in 2008 added consent requirements, and the HFEA's 'Consent for Disclosure' (CD) forms now cover the release of PII for research purposes.[14] The consent is divided into 'contact' and 'non-contact research'. 'Non-contact' asks that patients allow the use of their data for research, stating that PII may be disclosed for linkage purposes. Importantly, this consent applies to the patient and any child conceived.

This amendment has had a significant impact. When the CD forms were introduced just 30% of patients consented to the use of PII in (non-contact) research. By 2013, this had risen to about half of all patients, but opt-outs varied by clinic from less than 10% of patients to more than 90% (HFEA staff, written communication, 24 July 2013). A significant proportion of patients continue to opt out nationally: based on their initial registration forms, in 2018 44% of patients refused to allow future contact for research, and 30% did not consent to the use of their data for non-contact research (HFEA staff, written communication, 18 January 2019). Low consent rates cast doubt on the validity of research conducted using HFEA register data recorded since 2009. Given the increasing use of ART, rapidly developing technologies and the speed at which innovations are adopted, as well as the limited epidemiological evidence available regarding the long-term outcomes for women and their children, these low consent rates represent a missed opportunity to address these questions using UK data.

The aim of this study is to understand how and why patients decide to consent to (or refuse) the disclosure of Personal Identifiable Information held by the HFEA for research and linkage.

## METHODS

### Interviews

In-depth semistructured interviews conducted between September 2015 and December 2017 explored the experiences of patients who had completed the HFEA 'Consent for Disclosure' forms. Study information was provided to patients in nine clinics across England, treating both NHS and self-funded patients, and also advertised online via fertility forums and social media. The recruitment aimed for a diverse sample of patients, with a range of fertility histories and diagnoses. Individuals who had received treatment in England in the past 5 years and who had completed the HFEA 'CD' forms relating to the release of PII from the HFEA registry for use in research were eligible. Patients could contact the study team for more information and ask questions before deciding whether to participate. Interviews were arranged at a time and place convenient to the participant, either in their own homes or at a private space nearby.

Couples were encouraged to be interviewed separately, but two men were only willing to take part if their partner was also present. All interviews were audio-recorded, transcribed verbatim, checked and anonymised prior to analysis. Participants were able to review the transcripts if they wished. Participants were asked for basic demographic details at the end of their interview including age, self-identified ethnicity, marital status and job title, which was then used to derive socioeconomic status using the National Statistics Socio-Economic Classification tool.[15]

Clinic staff involved in explaining or checking the forms with patients were also interviewed, following the same recruitment, consent and interview process. While the transcripts from the staff interviews were analysed separately from the patient interviews, the findings were interpreted as part of the wider context of consent issues.

Interview participants were recruited until no new themes were identified in the transcripts, suggesting saturation was reached. Part of this study population are extremely difficult to recruit, namely individuals who opted not to share data because they did not want to take part in research or to divulge personal information. The online survey was added to try to maximise the responses from this group.

### Online anonymous patient survey

An anonymous online survey was added, as patients who did not wish to provide their personal identifiers or discuss sensitive issues may be more willing to contribute their views anonymously in writing, rather than face to face. The online survey included five simple questions employing a combination of drop-down menus and free-text answers; these are shown in table 1. This survey was advertised on fertility discussion boards (eg, fertilityfriends.co.uk), and promoted by fertility charities and support groups via social media (eg, Fertility Network UK). In addition, groups with a particular interest in data security and medical data were asked to publicise the study (eg, MedConfidential). The free-text responses were analysed alongside the interview transcripts.

### Analysis

The transcripts were anonymised and coded using Nvivo V.11 software.[16] The lead author (CC) read the transcripts repeatedly, and developed a coding framework for analysis in discussion with the coauthor (LH). This was an iterative process, which used a thematic approach identifying anticipated topics while also allowing new ones to emerge. Based on a modified grounded theory approach, the data were then analysed using the 'One Sheet Of Paper (OSOP)' method[17] which involves summarising all the coded extracts for a theme (including those that diverge from the consensus opinion) to ensure that nothing significant is omitted. Illustrative quotes are used in reporting the findings.

### Information governance

Individuals who agreed to interview gave signed, informed consent to participate after being provided with study information and the opportunity to discuss with the researchers. Copies of the participant information sheets and consent forms are available on the study website. Participants received information sheets, gave signed informed consent and were provided with a copy of the signed consent form for their records. No PII was collected on online participants, and as such consent was not required.

### Public and patient involvement

A study PPI group, comprising patients who had been through the experience of fertility treatment in England plus a patient advocate from a fertility-related charity, have had input into the development and design of this study. They also specifically commented on the design of the patient information leaflets and helped to revise the documentation and recruitment approach. Their input will be sought in disseminating the findings to a wider audience.

### RESULTS

Twenty patients were interviewed; 15 women and 5 men. All male interviewees were partners of female participants,

**Table 1** Questions used in the anonymous online survey

| Question | Response format |
|---|---|
| 1) When you started fertility treatment, you will have completed some consent forms. One of these was a 'Consent for Disclosure' form (for example, see https://www.hfea.gov.uk/media/2740/cd-form-v9-2-january-2019.pdf). Please tell us if you chose to allow your data to be used for **non-contact** research: | Drop-down menu: yes/no/can't remember |
| 2) Was it an easy decision? Can you tell us why you made this choice? | Drop-down menu: yes/no/can't remember Then: free-text box |
| 3) Please tell us about your experiences of being given this paperwork, and completing it. How did you get the form? Did you discuss it with anyone? | Free-text box |
| 4) Did it make any difference that this was about your fertility? (yes/no/can't remember box) Do you feel the same way about sharing other information about you? | Free-text box |
| 5) If you feel comfortable, please tell us about your fertility journey. | Free-text box |

Bold text signifies emphasis.

**Table 2** Characteristics of the (patient) interviewees

| | N (%) |
|---|---|
| Total patient interviews: | 20 (100%) |
| Women | 15 (75%) |
| Men | 5 (25%) |
| Ethnicity, white British | 16 (80%) |
| Age, median (range) | 36 (30–46) |
| Occupational social class* | |
| Managerial/professional occupations | 14 (70%) |
| Intermediate occupations | 2 (10%) |
| Routine and manual occupations | 3 (15%) |
| Student | 1 (5%) |
| Diagnosis | |
| Unexplained after investigations | 1 (5%) |
| Female factor (eg, anovulation, endometriosis, fibroids) | 6 (30%) |
| Male factor (eg, azoospermia, asthenozoospermia, cancer) | 4 (20%) |
| Both partners have a fertility diagnosis | 7 (35%) |
| None (single woman, or same-sex relationship) | 2 (10%) |
| Funding for treatment | |
| NHS-funded (ie, free at point of care) | 7 (35%) |
| Self -funded (ie, paying privately) | 7 (35%) |
| Experience of both (NHS-funded cycles, then self-funded) | 6 (30%) |
| Successful treatment (pregnant at interview or had a child using ART) | 10 (50%) |

*Individuals who are not currently working are classified by previous job title (n=2).
ART, assisted reproductive technology; NHS, National Health Service.

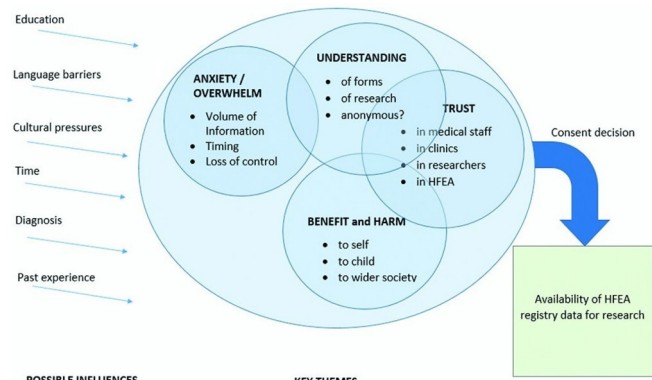

**Figure 1** The main themes, subthemes and influences affecting the decision to consent to data use. HFEA, Human Fertilisation and Embryology Authority .

one was interviewed with his partner present, and one couple was interviewed together. Fourteen respondents had agreed to allow their HFEA data to be used for research, two had refused, one had said both no and yes at different times and three were not sure. In addition 40 patients also completed the online survey (32=yes, 4=no, 2=yes and no at different times, 2=don't know/can't remember). Nine staff were interviewed, including doctors, nurses and administrative staff.

Among the patient respondents there was a range of fertility histories and diagnoses; table 2 shows the characteristics of the patient participants. Overall, interview participants were relatively affluent, with 15/20 reporting jobs titles indicating higher managerial, administrative or professional roles. Of those in the highest SES group 3/15 had refused to share their data and 2/15 were unsure about what they agreed to, whereas all respondents in the intermediate or routine and manual occupational groups had consented to share data. Socioeconomic status of the online respondents is unknown. A number of significant themes emerged from the interviews; these are shown in figure 1 and discussed further below. Table 3 summarises the issues highlighted by respondents which may influence the representativeness of the registry data available to researchers, while table 4 shows practical and process-related factors. Pseudonyms are used in reporting the data: pseudonyms used in this manuscript and each individual's consent decision are shown in table 5.

## Overwhelm and lack of control in treatment

The HFEA CD forms were completed within the wider context of fertility treatment. These initial clinic visits are information and paperwork heavy; patients are asked to learn about their bodies, their diagnosis (or lack of) and their forthcoming treatment. While many are well informed about their condition, others (like Nicola) arrive with little prior knowledge, 'The nurse… was quite shocked at how little we knew… we literally showed up wanting everything to be explained'. Patients are required to provide financial information, medical history and complete numerous legal forms including those related to storage of gametes and embryos, and what happens should one partner die. Several described feeling overwhelmed, both emotionally and in terms of the amount of information and paperwork that they received; Karen spoke of 'being bombarded', while Matt said, 'our heads were spinning'.

Some felt the situation was out of their control, which influenced their decision regarding the disclosure of data. Jodie spoke of 'containing' and 'restricting' the use of her information, saying: 'In a process where you feel that you have no choice and no control I am not prepared to give that bit up'. This also led to prioritisation; as Kimberley says: 'the whole process has just been paperwork and signing things and they (the forms) just lose their importance …'. Staff seemed very aware of this issue, as Stephen (staff) explained ''it might feel like there is just a lot of paperwork to be done… We don't want them to think that they have to tick these boxes in order to get on… Which is why we give them a fairly long appointment, and back it up with a nurse appointment as well'.

**Table 3** Factors that may influence the representativeness of the data available to researchers

| Characteristic influencing response | Exemplar quotes |
|---|---|
| **Educational attainment and literacy:** Where patients do not understand the forms or the point of disclosure they are less likely to give consent to disclose PII. | **Mark (staff):** 'Some patients they say that if they do not understand or they are hesitant, they just tick, no, no, no.' |
| **Language barriers:** For patients who do not speak English, staff focus on explaining treatment and have little time to devote to explaining the consent for disclosure forms. | **Barbara (staff):** 'we do have a lot of people who we need to have interpreters or we need to use language line with and that can be quite stressful for the patient because they have got a stranger in the room talking about their very personal business and in fact I had a very lovely couple but he was, because of his religion he had great difficulty using words like sperm and egg. It was very hard. To be honest with you, we don't probably talk about research in that situation because it would be too difficult and confusing for them, I think.' |
| **Cultural attitudes to infertility:** Differences in uptake across some BME groups; men may be less likely to agree to disclose their information than women. | **Valerie (staff):** '(British Asians) are very protective about their confidentiality and they always think that it is going to link in even if we are saying, this is just data, and it is numbers. I find them very much more private.' **Nicola (patient):** 'I think it stems from this very basic fact, in the beginning that conception is normally a sexual act and it is something very private and you don't normally discuss that with your grandmother and your aunt and everybody else, you just announce the happy news and everyone pretends they don't know how it happens.' |
| **Individual clinics:** Consent rates vary by clinic. Some spend more time and effort explaining the forms (eg, text message prompts to complete clinic-specific information sheets). Clinics where research is part of their culture may have better rates. | **Mark (staff):** 'Our nurse co-ordination time is 1 hour… Now when I was at the NHS, they (had) group co-ordination. Ten couples do one night to discuss about the treatment and things and then they will be seen individually after the co-ordination, so the individual appointment will be just like 15 minutes.' |
| **NHS versus private setting**: NHS patients may not get as much time with staff as private patients, and in some settings they have initial nurse consultations (to explain timing, drugs and forms) as a group. Self-funded patients do not even have to tell their GP that they are receiving ART, and may also be less likely to disclose PII for research. | **Alex:** 'we were so grateful to have the opportunity to receive IVF treatment through the NHS. We almost felt that we were helping research by providing data.' **Mark (staff):** 'Because this is a private clinic, most of the patients don't want to disclose their information to their GP…. you just have to explain why do I have to give information to my GP. That is the most common question. Why do I have to tick this?' |
| **Diagnosis:** | **Stephen (staff):** 'The thing is the patients that want to keep it more private tend to also want to put restrictions on other things as well. I mean I would say, it's probably a bit more likely (to refuse consent) if they are using donor gametes.' |
| **Successful outcome of treatment:** Can work both ways. Refusing consent after treatment could be important if opt-outs are applied retrospectively. | **Jude:** 'The first time I agreed because I believe in the value of sharing information for research purposes. However, the second time I completed the forms, I had a child (born from successful first treatment) and I did not feel happy with the clause that noted agreeing to disclose information meant that my child's identifying information would automatically be disclosed as well. It felt inappropriate and intrusive to my child's privacy. After much agonising, I refused, even though I regretted having to hold back my own information.' **Karen**, who completed the forms more than once, described how she had changed her mind: 'I don't think that you fully appreciate what is being asked of you…We have a son, we have benefited from other people participating in research. So, it put a different perspective on it as we could see why it was important to actually be included in the future.' |

ART, assisted reproductive technology; BME, black and minority ethnic; GP, general practitioner; IVF, in vitro fertilisation; NHS, National Health Service; PII, patient identifiable information.
Bold text signifies emphasis on the issue identified and the individuals who are quoted to support the findings.

## Benefit and harm

The benefits of research were raised by a number of respondents. Some considered sharing data for research was for 'the greater good', as Becca described it: 'a typical altruistic kind of thing, to do something to help women or more generally in the future,' while Jane consented 'mainly because I think it is a 'good thing'. Like picking up litter.' In some cases, a personal benefit was described.

Becca said that 'selfishly there might be some actual direct benefit to me'. Zoe took a longer-term perspective: 'even if it wasn't going to be successful for us, the research … might help so by the time (our son) is having his family, IVF might be 99% and that would be amazing'.

Some raised potential harms arising from sharing personal data such as the risk of fraud, identity theft and being targeted for marketing purposes. Karen thought

**Table 4** Practical and process-related recommendations arising from this research

| Issue raised | Exemplar quote | Recommendations from this research |
|---|---|---|
| The CD forms are difficult to understand | **Suzanne:** 'So I think your average person reading the form would go, I have no idea what that is… I am a teacher and university-educated, I don't know whether the wording is a little bit too academic. So I got a gist, even I had a rudimentary understanding of what it was.' | ✓Use **plain English and avoid technical jargon.** ✓Explain how data are protected and **who is accountable** for the Registry information. |
| People see them as unimportant | **Kimberley** says: "Because you are kind of just filling out forms all the time. They just kind of lose their importance which sounds a terrible thing to say but they do ultimately lose their importance. | ✓Provide examples of the **types of studies that may be done, and why they are needed.** ✓Clarify that the HFEA CD forms differ from other research consent (eg, trials) administrative paperwork (eg, finance), or consent forms for current treatment/storage. |
| Experience of receiving, completing and discussing the forms varies markedly by clinic | **Francis** (consented): 'The nurse explained what the paperwork was for and told us to take it home and read it and then fill it in next time we went, at the second visit, she then went through it again and we signed it.' **Pat** (consented): 'Discussed with senior lead in clinic what it meant, types of usage and what is used.' **Lee** (refused): 'I was passed the paperwork by the receptionist. None of it was discussed with me.' | ✓Patients benefit from having **time to read and consider** the paperwork in advance (a week or more). ✓An **SMS/text reminder** to complete the forms prior to their appointment may improve completion rates. ✓**Clinical (rather than administrative) staff** should check the forms and discuss patient queries. |
| Patients cannot recall what they signed | **Karen** (patient): 'They don't give you copies of the forms that you have signed. You never see them again whereas normally anything you sign, the bank or whatever, you get a copy of what you signed. They don't do that, so I could not even refresh my memory about what I signed.' **Valerie** (staff): 'All our forms that we get are carboned. The HFEA don't say that we have to give them a copy but that is a decision we took because we think it is best practice really.' | ✓**Provide patients with a copy** of these forms for their records (some, but not all, clinics do this already). |
| When staff are unclear about the paperwork, they do not always get adequate support | **Nina** (patient): 'The nurses do not know any more than what is on the form and so they will look at the form and go 'Oh, well' and say it again and then you (think) 'oh, well, I'll just tick a box'. I don't think that they know and I don't suspect that they are that interested.' **Hayley** (staff): 'When we have conferences, we will often have an HFEA representative who will come in and will say this is what is changing and this is what is happening.' **Michelle** (staff): 'Sometimes, I think that the forms are hard for the patients to interpret, let alone… and I have been on the HFEA website and it doesn't really give you any more information.' **Michelle** (staff): 'Sometimes it can be quite scary to contact the HFEA because they are the regulator and it is usually only the consultants and the senior nurses who have any contact with them, so they can seem for me quite distant.' | ✓**Training and support** should be provided for the staff who are explaining consent and checking the forms. ✓Provide a **contact person to answer queries** at the HFEA (or provide FAQs for staff). |
| Potential for completed forms to be inadvertently given to other patients, breaching confidentiality | **Valerie** (staff): '(I think that the forms should) have the patient's name on the front. I think that was a really bad thing… So when you pick up that form though, we have had before, not very often but not that long ago, someone had filled it in, left it in an outpatient room, then a healthcare assistant was tidying up thought it was an empty form, stuck it in a consent pack, do you know what I mean? I think that it needs to be quite clear that this is somebody's form.' | ✓Revise the form to **include something on the front cover** which clearly indicates completion. |
| Concern about consenting for a 'future' child | **Michelle** (staff): 'The couple who wanted Yes for the research and then said No for anything after, I know for him it was about……(the child)…. even though she wasn't pregnant at that point.' | ✓Consider separating the patient and child consents, or providing a better explanation of why both are required. |

CD, consent for disclosure; FAQ, frequently asked questions; HFEA, Human Fertilisation and Embryology Authority; SMS, short message service.
Bold text signifies emphasis.

that 'maybe (it) need[s] to be clearer on the form that none of the information you provide will be used for marketing purposes, it will only be used for research'.

Where anticipated harm was due specifically to the sharing of fertility data, participants worried about unintended effects for themselves or their children. Valerie,

**Table 5** Pseudonyms and consent decisions (note: only online respondents with quotes included in the main text are listed here)

| Pseudonym | Staff/patient | Did they consent to share HFEA data for research? | Interview or online survey |
|---|---|---|---|
| Alison | Patient | Yes | Interview |
| Amanda | Patient | Yes | Interview |
| Becca | Patient | Yes | Interview |
| David | Patient | Yes | Interview |
| Dominic | Patient | Yes | Interview |
| James | Patient | Yes | Interview |
| Jane | Patient | Yes | Interview |
| Jodie | Patient | No | Interview |
| Karen | Patient | Yes and no at different times | Interview |
| Kimberley | Patient | Yes | Interview |
| Leyna | Patient | Unsure | Interview |
| Matt | Patient | No | Interview |
| Natalie | Patient | Yes | Interview |
| Nicola | Patient | Yes | Interview |
| Nina | Patient | Yes | Interview |
| Sally | Patient | Yes | Interview |
| Stephanie | Patient | Yes | Interview |
| Suzanne | Patient | Unsure | Interview |
| Tim | Patient | Unsure | Interview |
| Zoe | Patient | Yes | Interview |
| Alex | Patient | Yes | Online |
| Bea | Patient | Yes | Online |
| Francis | Patient | Yes | Online |
| Jude | Patient | Yes and no at different times | Online |
| Laurie | Patient | Yes | Online |
| Lee | Patient | No | Online |
| Pat | Patient | Yes | Online |
| Sam | Patient | No | Online |
| Terry | Patient | Yes | Online |
| Abbie | Staff | - | Interview |
| Barbara | Staff | - | Interview |
| Hayley | Staff | - | Interview |
| Mark | Staff | - | Interview |
| Michael | Staff | - | Interview |
| Michelle | Staff | - | Interview |
| Sarah | Staff | - | Interview |
| Stephen | Staff | - | Interview |
| Valerie | Staff | - | Interview |

HFEA, Human Fertilisation and Embryology Authority.

an experienced research nurse, said that she thought patients refused consent *'because they just don't want anybody to know about it'*. Amanda explained why she did not tell anyone, saying: *'I was just protecting myself'*. **Jude** (online respondent) was unhappy that 'agreeing to disclose my information meant that my child's identifying information would automatically be disclosed as well. It felt intrusive to my child's privacy'. Matt explained that 'When I was growing up there was a big stigma about test tube babies'. His partner, Nina, expressed similar concerns when interviewed separately: 'I just don't think it is right, particularly if he is going to grow up with people knowing stuff about him that he doesn't know about himself. What if someone tells him? That is the type of stuff which pulls families

apart. …Ultimately, it will be a massive web of secrecy and the HFEA bit has just got tagged into it as well.'

Yet, for many, it was impossible to see a downside to data sharing. For Steph 'it was just a no-brainer, if for the sake of ticking a few boxes, it's not going to be detrimental to you in anyway, but beneficial to other people. For others, like Zoe, the fact that it was specifically health data was important: 'I kind of can't see how health information would be of use to anybody else apart from for a good cause'.

### Understanding

There was substantial variation in the level of understanding, with anonymity being a particular stumbling point. The data were often described as 'anonymous' when in fact the consent is to allow sharing of identifiable data. When asked what data the HFEA hold on her Nina replied 'I don't really know but I know that it will be anonymised so in that respect, I don't care'. Others, like Suzanne, thought that the data would be aggregate: 'it would be disclosing information on the area that we live in, the age bracket… It would be generalisations'.

There was also a common misconception that medical data are already easily available to researchers, as when Jane asked: 'Can't you just get the stuff off the NHS anyway?… Wasn't that a huge thing when they came up with electronic records and people were complaining and you can opt out of that if you want to? But you can't get it?'. However, some patients were very clear about what they had agreed to. Nicola knew 'that they would know our full name, our date of birth, our NHS number and by consequence that gives them information about our medical records'. For Jodie her understanding regarding the lack of anonymity was the underlying reason for the decision not to consent: 'the information was patient identifiable because they wanted the NHS number. … you know, potentially I think the rules and guidelines governing research could always change in the future so therefore using my NHS number, could link me to my maternity notes, you could link me to my dental notes, if I contact with the mental health services…'.

While staff had a good understanding of the paperwork, some misconceptions persisted. Hayley (staff) described how she (incorrectly) reassured patients who were concerned about data security: 'They think it is not going to be secure and that anyone can take that information. But I say, you will become a number, you are anonymised completely'. Similarly, Valerie (staff) noted that she tells patients 'They will be non-identifiable. It is just data which goes into a big data pot'.

Michael, a staff member, commented that 'I think that we need to re-evaluate the necessity of the degree of privacy. Twenty-eight years down the line [after the 1991 HFEA Act] I think that IVF is much more the norm'. He felt that 'we could adopt the standard NHS rules of engagement in terms of confidentiality, and I don't see why the HFEA Act should make it more complicated'. While all staff must comply with confidentiality requirements, for Michael there was a concern that 'because of the HFEA Act (a breach of confidentiality) is a more potentially serious breach because it breaks the law'. (In actuality, a breach of confidentiality would still break the law, under the UK Data Protection Act.[18]) This suggests that the perceived legal implications of working with HFEA data are a source of concern for staff.

### Trust (or lack of)

Trust was a strong theme acting on a number of levels from individual (eg, partner), to clinic or hospital, to wider institutions (the NHS or HFEA) and even broadly at the societal level.

#### In partner

Often partners filled in the forms together, or one completed them for her/his partner to sign. Stephen (staff) commented that 'the majority of the time the female partner has probably filled out the form and told him where he needs to sign…'. Suzanne explained how her partner asked her for guidance, and trusted her judgement: 'I think he did not want to admit that he did not understand. So, what did I understand it to mean and was I then happy to tick it?'.

#### In medical staff

Trust in clinic staff was also important. Patients noted their professionalism or warmth, and that rapport may help build trust at an emotional time. For example, Becca said of the doctor at her initial visit: 'she was beyond what I could have expected, she was very kind, very sympathetic'. Zoe's past experience influenced her trust in the medical teams: 'doctors, nurses have always looked after me and have made the decisions that have been right in the end for treatment, and you just trust them, don't you?'. However, even for Zoe this was not wholly positive: 'we put trust in their advice and we thought everyone else must do it so we thought, just sign'. Tim suggested the potential for (inadvertent) coercion: 'you do feel quite vulnerable in this position, you are putting a lot of trust and faith into this organisation to help you make a baby so sometimes you (are) answering the form in a way to please them, you know?'.

#### In NHS

There was a common misconception that it was the NHS data that were being shared. The NHS was generally seen as a trustworthy data holder, as Tim explained: 'our data is being used regardless, and I am a bit more trusting of the NHS to use my data'. However, this was not always the case, with an online respondent, Sam, noting that the reason for refusing to consent was a 'mistrust of NHS data use after recent scandals for example, care.data and the incoming 'data lakes''.

#### In the clinics

The idea that many fertility clinics are businesses, with financial motivations, negatively impacted on the trust that patients felt. Jane explained that 'we were going from state funded to private and we were quite anxious. The potential

for cowboy builders is bonkers'. James felt that 'if you want to be nasty about it, it is a baby making system and that is their end product'. For Amanda, this was a real source of concern: 'it can become a slippery slope. I have friends who have got into thousands and thousands of pounds of debt'. Even NHS-funded patients, like Kimberley, were required to complete finance forms since frozen embryo storage costs are only met by the clinical commissioning group for a short period. This highlights the transactional aspect of ART, which is often new to patients who have generally experienced 'free at point of delivery' care via the NHS. This situation is going to become more acute as commissioners continue to cut NHS fertility funding, forcing patients to self-fund treatment. Staff also discussed the 'business of IVF', highlighting that financial reward was not the main driver, as Hayley (staff) says: 'we are a business, but we are also in the business of trying to get women pregnant and the one thing that we want to do is hear a positive test, not 'oh no, it is negative' because that is just as miserable to us as it is to them'.

### In the HFEA

Legislation requires the HFEA to record information, and to ensure the welfare of any child born through treatment by checking the suitability of patients as parents, which led to some resentment of the regulatory body. Terry specifically highlighted this: '…fertile couples are not asked such questions before they're allowed to try to conceive'. Nina explained that '(the HFEA) demand so much information from you already, stuff that I would not necessarily want to give but you are over a barrel to give it…I feel a little bit resentful that when they want that extra money off us and they want all of this data, if we didn't have any fertility problems, we would not have to give information like this widely'. While likely a rare occurrence this attitude can have an impact on accessing care. Valerie (staff) recalled one patient: '(the patient) refused to complete the forms at all because he would not be registered with the HFEA, he felt they were like big brother, he wouldn't (sign the forms) so we couldn't treat him'.

Patients are aware that IVF is often unsuccessful. The publication of clinic success rates is intended to bring transparency to the industry, but patients do not understand how to interpret them. Becca felt like she 'had to trust (the HFEA) statistics because if I could not there was nothing I could trust', but Jane did not feel the same, explaining 'I am sure that they are true, they don't just make them up, but it is massaged and the HFEA gives you live births versus chemical pregnancies…'. Amanda also brought up this concern: 'that live pregnancy is only a fetal heartbeat found at the first scan… and then one in five go on to miscarry before 12 weeks''.

Respondents also noted how little they knew about HFEA. Amanda would have liked more information: 'There was not leaflet in here (explaining) what they are, what they do. Yeah, ok they say their rules and regulations but you don't know if they are tied to drug companies, you don't know who they are'. Jodie felt that 'the HFEA and researchers you are very

detached from you. I have a sense of I haven't met them, I don't have any trust in them. It just feels like a big unknown'.

### In 'experts' and researchers

While 'experts' were seen as trustworthy, what patients gleaned from expert comments was often negative and raised concern about the quality of data, research and its interpretation. As outlined above, some patients start treatment with very little understanding of the process, others however are extremely well informed.

Jodie explained that accountability in research was really important to her: 'I think that (for) non-contact (research), it is that idea of slight paranoia creeps in, of minions kind of running around with my data doing stuff which I don't know about'. This was echoed by Michelle (staff): 'I know it is confidential but 'researchers' - you cannot put a face on to those people can you? So it seems like an organisation that is going to get all of your information'. When Amanda pointed out that '(researchers) are not going to publish (names) in a medical journal…', her husband, James, responded with 'Well you would hope not, unless somebody hacked it or found it or found it in a bin bag or on a laptop or whatever they do on the train. Well it has been done…'.

### In broader society

For Tim, there was a sense that his data is already 'out there'. When asked who might hold his PII information Tim listed 'Banks, work, marketing agencies, tax man, any company that I have bought something that I have had to give my name on. The freebie websites I sign up to. It is more than I would actually know as well as GPs, hospitals, usual stuff as well, you know, passport people, biometrics, it is massive…. do I have any concerns? I used to but not anymore. It is out there so you can't do much about it'. James expressed a similarly resigned view: 'somebody will find it if they are clever enough and they wanted it that badly'. Both Tim and James had consented to disclose their PII.

Jane noted the influence of the media on her opinion of data use more generally, but went on to highlight the distinction she felt about the use of information on the fertility register: 'I guess I am a classic middle class Guardian reader about Google analysing my emails and things like that but I think when it comes civil service data collection it is often such a powerful and rich set of data because it is not collected for a research purpose, it is bureaucracy and therefore when someone comes along later and goes 'I really want to know about that' you can dig into that data and find out something that no one knew they were collecting information on. So, I think that agreeing to be part of that dataset opens up potential for people in the future'.

## DISCUSSION

While most respondents had consented to the disclosure of PII for research and linkage, it was clear that their understanding of what they had agreed to varied considerably. A common misconception was that the data were anonymous. While the benefits of sharing their data were generally seen

as quite removed from the individual, the potential harms described were more personal—either in terms of identity theft, hassle from marketing or the consequences of their ART becoming public. Trust was an important theme, which acted on a number of different levels. For some, this was tempered by a sense of resignation regarding how much control they actually have over their PII or health data. In addition to the emergent themes, a number of characteristics that may also be associated with consent rates were discussed, which are reflective of the existing literature on patterns of consent for data use[6] (see table 3)—if correct, this brings into question whether those who consented can be considered representative of the population of patients undergoing ART in the UK. Patients and staff also highlighted a number of process-related factors that may influence the decision to consent, and which vary between clinics and could perhaps be standardised (see table 4 for a summary of recommendations arising from this study).

## Comparison with existing literature

Previous studies have identified ART as a stressful and emotional experience and a perceived lack of control during treatment, as seen in these findings, is common. The overwhelming quantity of information and paperwork is also consistently reported by ART patients (for example, in interviews on HealthTalk[19]). It is important to remember this is the context in which patients are completing all CD paperwork. Often they have been waiting months or years to start treatment. These early interactions with the clinic are filled with anxiety and expectation.[20] Research in cognitive psychology has shown that people are rapidly overwhelmed by having more than a few options when making choices—these forms are completed alongside making many other decisions.[21] The CD was not always a considered decision; instead it was completed rapidly and later patients were unable to recall or explain what they had agreed to. Some patients may fall back on this 'tick box exercise' because they prioritise the other forms (treatment information and consent regarding legal parenthood, and what happens to gametes or embryos).

Research guidance emphasises the need to allow patients to consider potential benefit and harm when choosing whether to take part in research,[22] yet this was not as significant a theme as anticipated in these interviews. Altruism and the concept of giving for the greater good have previously been identified as reasons for consenting to participate in research studies.[23] A personal benefit, as described by some participants, is also seen as justification for consent in some studies but this is generally seen in intervention studies where there is a possibility that participants will be in the treatment arm.[24] The HFEA CD is unlikely to lead to direct personal benefit, since sharing of data presents less of an opportunity to benefit than other types of research.

Perceived harm fell into two domains—the 'lesser' harms relating to the sharing of PII, and the 'greater' harms of sharing information about fertility treatment. It is recognised that concern about fraud, identity theft and particularly the inconvenience of being targeted by marketing companies can prevent people participating in research that requires PII to be shared.[25] Here the pervasive idea that our personal information is already available to anyone who really wants it, and the resigned tone of comments regarding data security, suggest that this is not a particularly strong reason to refuse consent.

Harm to loved ones through disclosure of private information is more specific to fertility treatment. This has been observed previously in the context of divulging sensitive information relating to reproductive and sexual health,[5 26] and likely applies to discussion of paternity and the use of donor gametes. This appears to be closely tied to a perceived stigma around infertility[27] and a desire to protect the child and the parent/family unit from public awareness and judgement. There is also a sense of some patients protecting themselves from further upset by keeping information about their treatment private—meaning that they can choose to discuss unsuccessful treatment rather than being asked by family and friends. This is another way fertility patients appear to take some control of their situation, when so little of the treatment outcome is within their control.

Ethical research involving people is built on informed, voluntary and fair consent.[22] Yet, the level of understanding regarding data use, research and the permissions conferred by the consent form varied considerably. Recent reports have reflected the lack of awareness about research, data sharing and the use of patient records among the general public in the UK.[5] While studies have shown that the majority of patients feel that it is acceptable to use their anonymised health data, they would expect to be asked about the use of PII.[28] This is reflected in the ease with which those respondents who thought that their data were anonymised gave CD. This misunderstanding must be addressed.

The most pervasive theme throughout these interviews was that of trust, which influenced the consent decision in many ways. Where trust is lacking people tend to feel the need to retain control themselves; it has previously been reported that reduced trust in the body holding and processing information is associated with a desire to retain the option to consent or refuse data use.[5 29] Consistent with previous findings, trust in the NHS seemed reasonably high in most of our interviewees, while trust in academic researchers was more mixed.[30] It has been reported that trust is lowest in 'For profit' organisations, and that is also consistent with our findings where participants suspected private fertility clinics of ulterior motives for research or the publication of statistics. Some clinics may be unwilling to spend the additional time required to properly explain about the various ways in which the data may be used and to take the consent. However, comments made in this study about the potential legal issues arising from an accidental disclosure indicated that uncertainty over legal ramifications of keeping, storing and disclosing data may be a greater influence on the way that staff deal with consent discussions. There may be an additional concern that the clinic would be at risk of a complaint or possibly even litigation from the party or parties involved should an error be made in recording or reporting consent. This heightened awareness may influence the way

that staff explain the CD forms to patients, or mean that they discourage consent to data use.

The HFEA was more strongly associated with the 'for profit' IVF industry than with NHS clinics, when in fact they regulate all treatment (NHS and privately funded). The lack of awareness among participants about the role and intentions of the HFEA can only be detrimental. In terms of consent, a recent Open Data Institute report found 64% would share some personal data with an organisation they know, compared with just 36% for an organisation they don't,[30] which highlights the need to improve the public's knowledge and understanding of the HFEA and trust in their oversight.

## Strengths and limitations

The researcher/interviewer was aware from the start of the sensitive nature of infertility, and initial interviews started with questions about the 'less sensitive' topic of data sharing. However, the researcher reflected on how the interviews may be influenced by the participants' perception of her expectations, as well as her own preconceptions. In particular, she was concerned that interviewees would assume that as a 'researcher' she must be in favour of data sharing, and that this may colour their responses. To try to minimise this, the interviews were reorganised, so that the opening question asked patient interviewees to share their 'fertility story', allowing them to outline their experiences and highlight what had been significant for them. In addition, the second researcher assessing the transcripts and themes allowed another perspective on the analysis.

Recruitment was challenging, and the majority of participants had consented to the use of their HFEA data for research. This is unsurprising, since the nature of the study meant that we were trying to recruit patients who had chosen not to consent to other research. Patients contacted via clinics received information about this study early in their treatment, and a reasonable number arranged interviews but then withdrew when treatment started or their treatment cycle failed. A diverse sample was recruited to garner a range of experiences but it is important to note that people of more affluent socioeconomic status were over-represented in the interview sample (in part, this may also be a reflection that many people have to pay to access fertility treatment privately). The online survey was added to increase input from patients who wished to remain anonymous; sociodemographic information is not available for these participants. The use of qualitative research methods afforded a richer data set than could be collected via a structured survey or questionnaire alone.

There is an ever-developing story about data security, use and consent in the UK, and these experiences and opinions were formed during the study period, 2015–2017. Additional influences on the decision to consent to disclosure of data may now also be present, such as increasing news stories about 'data harvesting'. In this environment, the theme of 'trust' that came out so strongly here will likely be increasingly important and relevant to how people approach the use of their data.[31]

## These findings in light of changes to information governance in the UK

The changing climate of information governance and consent requirements, such as the 2018 Europe-wide General Data Protection Regulations, mean that HFEA CD forms were revised in January 2019.[32] This was an opportunity make them clearer for patients, as it is evident that many patients did not properly understand what is currently being asked of them. The forms must be clearly written, in plain English and with minimal technical jargon. They should explain what types of studies can be performed, patients should be given time to consider the issues before completion and should be provided with a copy once completed (our recommendations are outlined in table 4). The HFEA has recognised the need to support clinic staff who discuss the forms with patients, and new written information for staff is available online.[33]

While it is the patients' right to decide whether their data can be shared, the failure to convey the benefits of allowing data to be disclosed for research use and the minimal risks of consent to non-contact research, means that patients who may actually be happy to share data opt not to. This cautious response, of refusing data sharing because the implications of what is being asked are unclear or seem irrelevant, does not represent true informed consent; it is the flip side of agreeing to something without full understanding. As researchers and clinicians we are at pains to ensure those who participate are fully aware of what they consent to, but it is just as important to ensure that those who opt out also do so with full comprehension. It is a legal and ethical requirement that patients make an informed decision, are clear what the implications are of their choice and what they can do if they change their mind in the future.

## CONCLUSION

Patients need to make many decisions at the start of their fertility treatment, often asking that they consider an 'unknowable future'—Will they ever have a child? Will they have unused embryos? Will the decisions that they make now actually be relevant in the future? Understanding what is being asked, what the potential harms and benefits are, and trust in those who will be keeping and using the data emerged as important themes. Time to read and understand the research consent paperwork, and timing of receiving the forms, are both important: while these questions come at the start of treatment, it may be years into a 'fertility journey' and during a period that is both exciting and stressful. Unlike other routinely collected data, the decisions are influenced by the perceived stigma and secrecy surrounding fertility problems, as well as wider concerns in the UK about data sharing and data security. Improving the clarity of the consent forms, providing better examples of how the data may be used and what benefits could arise, and building trust in the HFEA through improved outreach and information would give fertility patients the opportunity to make a truly informed decision regarding CD of personal data for fertility research. These interviews have pointed towards

potential sources of bias resulting from the self-selecting group who choose to opt-in to data sharing and linkage. Further work exploring the impact on the use of the registry data for epidemiological research is needed. As an ongoing legal basis for linkage and research using population registry data the reliance on consent in the longer term is untenable; alternatives are needed that are acceptable to patients and the public, and provide the necessary data for meaningful, informative health research.

**Acknowledgements** The authors thank the participants, and the clinics and staff who facilitated the recruitment for this study, including: Hewitt Fertility Centre, Liverpool Women's NHS Foundation Trust, Liverpool; Homerton Fertility Centre, Homerton University Hospital, London; Nurture Fertility, part of The Fertility Partnership, Nottingham; Mr Stuart Lavery, Consultant Gynaecologist, and his teams at the Hammersmith Hospital and Boston Place Clinic; and others.

**Contributors** The 'Taking pART' study was conceived and designed by CC (Principal Investigator), with input from JK, LH and MQ. CC and LH developed the interview strategy, CC conducted all interviews, CC and LH reviewed transcripts, and analysed the findings. CC drafted the manuscript with inputs from all authors.

**Funding** This study was funded by the Medical Research Council (UK) as part of a Career Development Award to CC (ref: MR/L019671/1). CC (as lead author) affirms that this manuscript is an honest, accurate, and transparent account of the study being reported; that no important aspects of the study have been omitted; and that any discrepancies from the study as planned have been explained.

**Competing interests** None declared.

**Patient consent for publication** Not required.

**Ethics approval** Ethical approval for the study was granted by London City & East Research Ethics Committee (15/LO/1305) and local site R&D with oversight for the NHS clinics.

**Provenance and peer review** Not commissioned; externally peer reviewed.

**Data sharing statement** The signed consent allows the use of interview transcripts for this study, but not for further sharing. As such transcripts are not currently available for secondary use.

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
