## [Reviewer comments · BMJ Open]

ARTICLE DETAILS

TITLE (PROVISIONAL)	"I haven't met them, I don't have any trust in them. It just feels like a big unknown": A qualitative study exploring the determinants of consent to use Human Fertilisation and Embryology Authority registry data in research
AUTHORS	Carson, Claire; Hinton, Lisa; Kurinczuk, Jenny; Quigley, Maria

VERSION 1 - REVIEW

REVIEWER	Kerina Jones Swansea University, Wales
REVIEW RETURNED	23-Oct-2018

GENERAL COMMENTS	Many thanks for the opportunity to review this manuscript: "I haven't met them, I don't have any trust in them. It just feels like a big unknown": A qualitative study exploring the determinants of consent to use Human Fertilisation and Embryology Authority registry data in research. It addresses an important topic in seeking to understand the determinants of consent to use HFEA register data in research. I have some major concerns with the paper as it stands and these would need to be addressed before I would consider it ready for publication. The paper focuses overmuch on the use of patient data in general and too little on the use of data specific to HFEA. As such, it misses an important opportunity. There are some phrases and terms used that show a lack of understanding of the area: e.g. data ownership; educating the public. There are also instances of speculation that don't appear justified. Some of the statements in the discussion don't appear to be informed by the findings. Some are sweeping. It would have been useful to see the questionnaire. There seems to be some conflation of issues around data sharing and obtaining the IVF. The paper contains a lot of quotes but lacks clarity on how the information was used to lead to the conclusions, and the summaries given don't necessarily follow from the points made.
---

	The themes identified are largely non-specific – they apply to data use in general. Again, a missed opportunity to focus on the topic at hand. The sample is small (ok, this is generally the case with qualitative studies) but it is also non-independent in that it includes some of the women's partners. But it is not clear how this has been handled or how it has influenced the findings. Some of the references are repeated e.g. ref 5 = ref 24. There is a lack of reference to data protection legislation. It is unclear how the themes in Table 2 have been derived. In summary, this could be a very useful paper but it needs a lot of work. This includes both context and content.
--	---

REVIEWER	Raywat Deonandan University of Ottawa, Canada
REVIEW RETURNED	19-Nov-2018

GENERAL COMMENTS	This is a nicely written and well packaged study that has implications not just for fertility research, but frankly for big data studies in general. Mind you, I have checked "insufficient references" in my review because several statements made in the paper have no references. I list below specific instances of where I require further information: line 82: It would be useful to include a one sentence summary of the mishandling of Care.data for we non-UK people who are unaware of this scandal. line 107: it is stated that 1 in 6 couples have problems conceiving reference? line 121: reference for the claim that 30% of patients consented? is the same reference as line 123? line 124: what is a significant proportion? reference? In terms of the results, demographic information of respondents is lacking. I would like to know the SES of interviewees, in particular, as this affects their penchant for disclosure and participation. As well, I'm unsure that thematic saturation was achieved. I would appreciate some comments on this issue, to ensure that the qualitative method was rigorous. Thank you.
---

REVIEWER	Lisa Eckstein University of Tasmania, Faculty of Law, Australia
REVIEW RETURNED	29-Nov-2018

GENERAL COMMENTS	In my view, this is an excellent contribution to the literature. It comes at a time of increasing scrutiny of waivers of consent for research uses of personal data and requirements to seek express consent for such use. Yet refusals of consent can jeopardise potentially beneficial research. Understanding the reasons for non-consent, as well as the authenticity of choice for those who consent to use or disclosure of their health data is paramount. This study was conducted in a particularly vulnerable patient group whose experiences of consenting, or refusing consent, to the use of their health information is valuable in its own right as well as an example of these kinds of decisions more generally. The methodology appeared sound and the resultant interview themes were clear and well developed. It would have been nice to have a few more "no" respondents but given the challenges of recruiting this group, the mix of participants was reasonable. In discussing the results, the authors did a good job of contextualising the responses provided, including, where necessary, correcting any misconceptions in a respectful way. As a very minor point, the third paragraph of the introduction had a few sentences that could use a proof (a couple of words seem to be missing). I also wonder whether it would be helpful to have a table linking the pseudonyms with their agreement or non-agreement to release their data.
---

VERSION 1 – AUTHOR RESPONSE

Authors' response to Reviewers' Comments:

Reviewer: 1

Reviewer Name: Kerina Jones

Many thanks for the opportunity to review this manuscript: "I haven't met them, I don't have any trust in them. It just feels like a big unknown": A qualitative study exploring the determinants of consent to use Human Fertilisation and Embryology Authority registry data in research. It addresses an important topic in seeking to understand the determinants of consent to use HFEA register data in research. I have some major concerns with the paper as it stands and these would need to be addressed before I would consider it ready for publication.

Thank you for your time spent reviewing this manuscript. We have tried to address each of the points you raise below.

The paper focuses overmuch on the use of patient data in general and too little on the use of data specific to HFEA. As such, it misses an important opportunity.

The research was conceived as a study to address the use of the HFEA data, but the authors believe that the themes that arose from the interviews are more widely applicable to the discussion around and understanding of public perceptions of data use. We do include discussion specific to the consent to use HFEA data where appropriate – for example, in the discussions around trust in the

clinics, the NHS and those using the data, and also Tables 2 and 3 which highlights characteristics perceived to influence consent and recommendations specific to HFEA records. Reviewers 2 and 3 both appear to be satisfied with the wider discussion of the findings, which we hope will be of interest to a broader readership, and so we have respectfully chosen to keep the wider discussion in the manuscript.

There are some phrases and terms used that show a lack of understanding of the area: e.g. data ownership; educating the public. There are also instances of speculation that don't appear justified. Some of the statements in the discussion don't appear to be informed by the findings. Some are sweeping.

We have given a reference for data ownership as an issue of debate – because whether or not patients actually 'own' their information, this has become a topic of debate in the wider discussions about the use of routine data. - Kostkova P, Brewer H, de Lusignan S, et al. Who Owns the Data? Open Data for Healthcare. *Front Public Health* 2016;4:7. doi: 10.3389/fpubh.2016.00007 [published Online First: 2016/03/01]

We have rephrased the sentences including the term 'educating the public' (see line 90), since the stated objective of Wellcome's Understanding Patient Data is to 'support conversations about data use' and we recognise that this need to improve understanding and awareness of appropriate data use applies to everyone.

We have reviewed the paper for 'instances of unjustified speculation'. We have identified one statement that was a remnant of a previous (longer) draft, which we have removed (see page 17), but we cannot find any other statements that we believe fit this description. If there are specific issues you would like us to address, please provide more detail.

We also want to emphasize that, by its very nature, qualitative research based on grounded theory is inductive, drawing interpretations of the data as emergent themes. This occurs through an iterative process of data collection and analysis, conducting interviews and coding with each stage informing the other. This recognition of themes within the data is at the core of what we do as qualitative researchers, and why taking a reflexive approach is so important; we hope that this is not being misinterpreted as 'unjustified speculation'.

It would have been useful to see the questionnaire.

This research used semi-structured interviews, and so there was no questionnaire. As is standard for this methodology, we developed a topic guide for both the patient and staff interviews, which can be provided to readers on request (attached, for information). This was iterative guide and it developed in response to the early interviews – for example, the first interviews focussed on data initially before moving on to questions about fertility history, as these were deemed more invasive and personal. However, it became clear early on that interviewees wanted to 'tell the story' of their fertility history and so this was moved to the start of the interview because it allowed participants to speak freely and encouraged them to speak about their experiences more openly.

We have now included the 5 questions used in the anonymous online survey in a new table (see new Table 5), so readers can see what was asked of those participants and the format of the responses that could be entered into the survey platform.

The themes identified are largely non-specific – they apply to data use in general. Again, a missed opportunity to focus on the topic at hand. There seems to be some conflation of issues around data sharing and obtaining the IVF.

The themes arose from the data, as is appropriate to this type of analysis. The individuals interviewed were talking about their consent to use data collected as part of their fertility treatment, but it was

clear that people think about issues around 'their data' more generally. Dr Jones correctly identifies the perception among some interviewees that there is a conflation of issues around data sharing and getting past the 'gate keepers' to undergo treatment. Rather than being a weakness, we believe it is useful to know this – it affects how patients and staff speaking patients need to be clearer about the separation of these issues.

The paper contains a lot of quotes but lacks clarity on how the information was used to lead to the conclusions, and the summaries given don't necessarily follow from the points made.

We are sorry that the methods used are unclear. The quotes selected are illustrative of the data that informs the themes, which we analysed using a modified-grounded theory approach. This is a standard methodology in qualitative research, and it is conventional to include such illustrative quotes. We cannot, of course, include all the data that inform the summaries, but hope that the quotes are useful in setting the context and 'hearing' the participants in their own words. If there are particular quotes/summaries that you would like us to address please identify these, as it is difficult to make changes to the text based on such a general comment.

The sample is small (ok, this is generally the case with qualitative studies) but it is also non-independent in that it includes some of the women's partners. But it is not clear how this has been handled or how it has influenced the findings.

The sample size is not small for qualitative research requiring in depth face to face interviews, and it was non-independent by design. Most people making a decision about the use of their HFEA data do so early in fertility treatment which they are undertaking as a couple. Although both partners complete their own copy of the Consent for Disclosure forms, this is usually done together or discussed. We interviewed men and women from the same couples for a number of reasons: first, as we were interested in how the consent for data use was discussed (or not) within a couple; second, because we wanted to see if one partner influenced the other; and third, because it was informative to hear the same 'story of treatment' from a male as well as female perspective.

Men and women could also volunteer to take part in the study without the involvement of their partners, should they wish. In the manuscript we have noted where we are quoting from partners. For example, on line 221 we quote Matt and Nina, who spoke about stigma, and on Line 338 where Amanda and James talk about data security. While it is clear from the existing text that Amanda and James were interviewed together, we have added to the manuscript to indicate that Matt and Nina raised similar concerns when interviewed separately. We have also clarified in the methods that while we encouraged partners to be interviewed separately, some men were only willing to take part if their partner was also present. Of course, it is not true that all ART patients undergo treatment in heterosexual couples, and so our interviewees include also women without male partners – either undergoing treatment as a single woman, or married to a woman – to ensure a broader range of experience was captured.

Some of the references are repeated e.g. ref 5 = ref 24.

The references have been checked and corrected where necessary.

There is a lack of reference to data protection legislation.

Apologies for this omission, it has now been rectified.

It is unclear how the themes in Table 2 have been derived.

The over-arching themes that came out of the analysis are presented in the main results. However, this cannot capture all of the very rich information that was available in the interview data. A number of specific issues were also raised by interviewees which highlight factors that may impact on

response. These are important for users of the HFEA data, highlighting factors which may influence the demographic characteristics of those patients who allow their data to be used in research. These are potential sources of bias in epidemiological research using the HFEA register data, which we believe need further investigation in quantitative analysis of the HFEA data.

Reviewer: 2

Reviewer Name: Raywat Deonandan

This is a nicely written and well packaged study that has implications not just for fertility research, but frankly for big data studies in general. Mind you, I have checked "insufficient references" in my review because several statements made in the paper have no references. I list below specific instances of where I require further information:

Thank you for your positive comment on the potential implications of the research. We have addressed each of your queries below.

line 82: It would be useful to include a one sentence summary of the mishandling of Care.data for we non-UK people who are unaware of this scandal.

I have now included some background information regarding Care.data. Please see paragraph 2 of the introduction, where the following text has been added:

Care.data was intended to make available linked primary and secondary care data for England to 'approved users' (including researchers and care commissioners) in pseudonymised datasets. A botched awareness campaign led to a public backlash about the lack of information, and concerns over data security and usage then put an end to what should have been an incredibly useful resource for health research and planning purposes.

line 107: it is stated that 1 in 6 couples have problems conceiving reference?

Citations have been added, referring to:

Oakley et al, Lifetime prevalence of infertility and infertility treatment in the UK: results from a population-based survey of reproduction. Hum Reprod 2008;23(2):447-50

te Velde ER et al, Variation in couple fecundity and time to pregnancy, an essential concept in human reproduction. Lancet 2000;355(9219):1928-9

line 121: reference for the claim that 30% of patients consented? is the same reference as line 123?

Citation has been added, referring to the same reference as line 123, which was a personal communication between HFEA staff and lead author.

line 124: what is a significant proportion? reference?

This referred to the same reference as line 123. However, we have now updated this based on a more recent personal communication with the HFEA to say:

A significant proportion of patients continue to opt-out nationally: based on their initial HFEA registration forms, in 2018 44% of patients refused to allow future contact for research, and 30% did not consent to the use of their data for non-contact research.

In terms of the results, demographic information of respondents is lacking. I would like to know the SES of interviewees, in particular, as this affects their penchant for disclosure and participation.

While we have information on job title which are indicative for SES, we cannot provide these for each pseudonym because in combination with other information (even just partner's job title, for example) they could allow the participants to be identified – which we will not permit. Instead, we have grouped these according to occupational social class and reported aggregate figures in Table 1. We have also added information on age to this table. A sentence has also been added to the methods to explain how socioeconomic status was derived.

It is unsurprising perhaps that the respondents tend to be relatively high SES, since it is a recognised predictor of participation in research generally. However, we have added a comment to this effect to the discussion.

As well, I'm unsure that thematic saturation was achieved. I would appreciate some comments on this issue, to ensure that the qualitative method was rigorous.

Sampling methods for quantitative and qualitative studies are quite different since they serve a different purpose – in quantitative research the aim is to sample individuals who represent the whole study population so that the results are generalizable, while sampling for qualitative research aims to recruit individuals to explore why people believe or behave as they do (for a helpful discussion see: Marshall MN. Sampling for qualitative research. *Family Practice*, 1996; 13: 522–525). There is no evidence that the attitudes, experiences, or values that inform qualitative investigation are normally distributed in the population, as such individuals cannot be selected randomly but instead are recruited to provide insight into the research question. In our study, we recruited via clinics (so that the interviews were near the time of making a decision about the consent forms, and also as a convenience sample given the rarity of ART) and via online fora and social media (to recruit individuals who felt that they could contribute to the research, perhaps because of their own experiences, and thus gaining a richer variation of experience).

Interviewing was conducted until no new themes emerged from the data. This is known in the qualitative literature as achieving 'data saturation' and is an accepted technique for studies that seek to reflect diversity. Overall, we believe that we have achieved reasonable data saturation. We would have liked to have interviewed more people who had refused to consent to allow their data to be shared (a particularly challenging group to recruit), but by including the anonymised online survey as well as interviewing the three people did not consent to data sharing we identified many influences affecting patient's consent decisions.

Reviewer: 3

Reviewer Name: Lisa Eckstein

In my view, this is an excellent contribution to the literature. It comes at a time of increasing scrutiny of waivers of consent for research uses of personal data and requirements to seek express consent for such use. Yet refusals of consent can jeopardise potentially beneficial research. Understanding the reasons for non-consent, as well as the authenticity of choice for those who consent to use or disclosure of their health data is paramount. This study was conducted in a particularly vulnerable patient group whose experiences of consenting, or refusing consent, to the use of their health information is valuable in its own right as well as an example of these kinds of decisions more generally. The methodology appeared sound and the resultant interview themes were clear and well developed. It would have been nice to have a few more "no" respondents but given the challenges of recruiting this group, the mix of participants was reasonable. In discussing the results, the authors did a good job of contextualising the responses provided, including, where necessary, correcting any misconceptions in a respectful way.

As a very minor point, the third paragraph of the introduction had a few sentences that could use a proof (a couple of words seem to be missing). I also wonder whether it would be helpful to have a table linking the pseudonyms with their agreement or non-agreement to release their data.

Thank you for your positive comments on this manuscript. We have revised the wording of the third paragraph, and checked the remaining text again for typographical errors.

As suggested, we now provide a table (Table 4) indicating the consent decisions for each pseudonym included in the text. This could be included as a supplementary table if the editors consider that more appropriate.

VERSION 2 – REVIEW

REVIEWER	Kerina Jones Swansea University, Wales, UK
REVIEW RETURNED	16-Feb-2019

GENERAL COMMENTS	The reviewer completed the checklist but made no further comments.
--

REVIEWER	Lisa Eckstein University of Tasmania, Australia
REVIEW RETURNED	28-Feb-2019

GENERAL COMMENTS	In my view, this revision has satisfactorily incorporated previous reviewer comments and is suitable for publication.
---